# Do Pre-service Teachers Dance with Wolves? Subject-Specific Teacher Professional Development in A Recent Biodiversity Conservation Issue

**Alexander Georg Büssing * , Maike Schleper and Susanne Menzel**

Department of Biology Didactics, Osnabrück University; 49074 Osnabrück, Germany;
mschleper@uni-osnabrueck.de (M.S.); susanne.menzel@biologie.uni-osnabrueck.de (S.M.)
* Correspondence: alexander.buessing@biologie.uni-osnabrueck.de

**Abstract:** Biodiversity conservation issues are adequate topics of Education for Sustainable Development (ESD), as they involve ecological, economic and social aspects. But teaching about these topics often challenges teachers due to high factual complexity but also because of additional affective dimensions. As a consequence, teacher professional development in ESD should address these affective components, to better qualify and motivate teachers to integrate conservation issues into their teaching. To investigate behaviourally relevant factors, we selected the context of natural remigration and conservation of the grey wolf (*Canis lupus*) in Germany and surveyed 120 pre-service biology teachers (M = 23.2 years, *SD* = 3.3 years) about contextual factors and their motivation to teach about the issue. Participants reported more positive attitudes, higher enjoyment and an increased perceived behavioural control towards teaching the issue in future teachers when they perceived a smaller psychological distance to the issue and an overall higher motivation to protect the species. As this motivation was grounded in more fundamental personality characteristics like wildlife values and attitudes towards wolves, we discuss the central role of these traits as a basis for transformative learning processes and the necessity of a holistic and subject-specific teacher professional development in ESD.

**Keywords:** teaching motivation; values; attitudes; psychological distance; return of the wolf

## 1. Introduction

### 1.1. The Return of the Wolf as a Context of Education for Sustainable Development (ESD)

Declining biodiversity is a major problem for sustaining life on planet earth [1]. But as the example of reintroduced wolves into Yellowstone National Park showed [2], there are ways to respond to declining biodiversity and recover ecosystems [3]. While the project in the Yellowstone National Park serves as an example of intentionally reintroducing a predator species into an ecosystem, wolf packs are also returning in natural ways to some of their ancestral habitats. Examples from Europe show the recovery of wolves in regions where they were eradicated through human activities in the last centuries [4]. However, this natural return often leads to conflicts between stakeholders, especially in densely populated regions [5].

The main conflict arises due to the economic damage experienced by livestock owners, which is a major cause for the rejection of wild wolves in affected areas [6]. In Lower Saxony, a federal state in the northwest of Germany, the issue of returning wolves is particularly controversial [7,8]. This region has the highest density of livestock in the country [9] and farmers are afraid of losing animals to predators when more wolves enter the region [10]. Besides the possible economic costs, the return of wolves also provokes opposition based on social factors like distrust, conflicting values or beliefs [11].

Furthermore, deeply rooted implicit beliefs and feelings led to the stereotype of the "Big Bad Wolf," which is promoted by society's subjectively perceived fears about the return of the wild animal [12]. These beliefs and fears are formed based on underlying values like wildlife value orientations [13].

While analyses from affected areas showed only a small proportion of livestock as a source of nutrition for wolves [8,14], the viable status of the species in new regions highly dependents on mutual solutions between the stakeholders [15,16]. Such solutions require an in-depth understanding of the conflicts and their management, which optimally leads to joint solutions considering the needs of all stakeholders [17].

The described interconnectedness of ecological (biodiversity), economic (the costs of livestock) and social dimensions (conflict between farmers and conservationists) underlines the suitability of the issue as a context of Education for Sustainable Development (ESD). In contrast to prior approaches of environmental education focused on sole ecological knowledge, ESD explicitly integrates these dimensions in education [18,19]. Prior studies showed conservation issues as a suitable way to address these interconnected dimensions with students [20] and foster their decision-making competencies [21]. As part of these decision-making competencies students should also reflect on their own values [22]. While this reflection is intended to develop students' sustainability competencies, school teachers are not allowed to overwhelm students with their opinions. This also includes the representation of all controversies surrounding topics like returning wolves. In Germany, this also became contractual in the consensus of Beutelsbach [23].

Nonetheless, teachers should address value reflection as a basis of such decision-making competencies in schools, as they are constituents of a wider general literacy [24]. Therefore, they also found their way into national curricula. For example, German regulations bind teachers to explicitly foster students' decision-making about biodiversity issues and foster these competencies in the subject of biology [22,25]. The issue of returning wolves has already been successfully utilized as a possible topic, linking biological learning opportunities, affective appeal and personal relevance [26].

## 1.2. Environmental Issues in Schools

Despite these normative preconditions to teach topics like the return of the wolf, teaching about such topics within the context of ESD remains difficult for teachers [27]. This might hinder teachers in the utilization of specific contexts, why we need to know more about the teaching motivation for teaching specific issues to properly design teacher professional development. One problem is the uncertainty due to the controversial nature of the issues [27]. Based on this controversial nature, a high amount of specialized knowledge is needed to make informed decisions about environmental issues [28]. Furthermore, teachers might have an unclear understanding of the intrinsically complex and abstract construct of biodiversity [29].

Besides these cognitive factors, worldviews may be another problem, as variables like values, experiences or contextual beliefs are inevitably linked to meaning- and decision-making in environmental issues [30,31]. This affects teaching, since a wide variety of personal or contextual variables may influence teaching habits and motivation in ESD contexts [32]. Other examples from climate change show how political views and teacher identity might influence science teachers in the United States to doubt human activities as a cause for the globally changing climate [33]. Especially in science subjects, this might lead to a concentration on factual knowledge instead of ethical decision-making [34,35]. Such teaching approaches neglect the high potential of conservation issues in order to foster ethical decision making in school students [36]. But there is only little knowledge about specific behaviourally relevant personality variables.

Further knowledge about such behaviourally relevant contextual values, beliefs and motivations may help to better understand pre-service teachers' learning in controversial issues and explain why discussions might become emotional [37]. With this understanding, teacher educators could try to focus more on the reflection of underlying values when discussing environmental issues, as intentional value shifts for the sake of conservation may not be possible due to the cultural nature of values [38].

Finally, a further understanding of values as part of teaching identity and motivation is important for successful professional development, as the effectiveness of development is aimed at engaging teachers with content instead of only offering it to them [39,40]. This knowledge is also needed to properly design teacher professional development [41].

*1.3. Teacher professional development in ESD*

Teacher professional development generally describes teachers' learning processes and seeks to understand how these learning processes can facilitate student learning [42]. Since early times, teacher development has concentrated on knowledge development [43,44]. Based on the differentiation of knowledge into the categories of content knowledge (CK), pedagogical knowledge (PK) and the subsequent pedagogical content knowledge (PCK), PCK in particular has been found to be a main contributor towards student achievement and learning outcomes [45]. Due to this importance, teacher knowledge emerged as one main facet of professional teachers [46]. But normative models like the model of professional action competence from Baumert and Kunter also propose motivation, consistent values/beliefs as well as self-regulative functions as other components of teacher's professional competence [47].

These more affective variables of teacher behaviour raise problems for teacher professional development, as they are at the core of understanding teacher behaviour but have been often neglected in prior research [40]. Due to the central role of emotions in ESD [48] as well as motivations and values as key competencies of sustainably literate citizens [49,50], there may be a urgent need in ESD to better understand teachers' beliefs and values and their influences on teaching [11,51,52].

Knowledge about such factors is also needed to adapt transformative learning to teacher professional development. Transformative learning aims at changing the learner's prior frames of reference to make them more inclusive, open and reflective [53]. For these learning processes, educators need knowledge about the prior way of thinking to transform this into more sustainable ways [54]. This especially concerns emotions and deeper personality factors as emotions and values [55,56]. Such professional development activities also have to respect the imperative of educational neutrality, similar to the aforementioned commandment of the consensus of Beutelsbach [23].

One concrete example of possible values for the context of returning wolves may be wildlife value orientations, which have predicted protection motivations in the context of returning wolves [57] and build the basis for higher order attitudes as well as motivations in wildlife contexts [13]. When wildlife issues should be included in schools, such values have to be reflected in initial teacher education, as teachers otherwise will rely on their everyday life values. In society, these values are often not coherent with the aims of ESD [38].

Besides these values, also teachers' general attitudes or the subsequent protection motivation towards wolves may influence teaching decisions and approaches when teaching about the species [58]. Furthermore, prior studies showed how other factors like the perceived closeness to the issue were connected to the motivation to protect the species [59,60]. Prior studies conceptualized this closeness either with the measured distance to the next wolf territory [61] or using psychological measures like psychological distance [59]. Due to their relevance for the protection motivation, these factors may also increase the teaching motivation based on a higher relevance of personally close contents [62]. But this increased closeness may also have reverse effects on the motivation to teach about the issue, as teachers may be inhibited by the regional political climate and could therefore try to avoid such topics [63].

*1.4. Aim of the Present Study*

To sum up, while values, beliefs and other non-cognitive variables like the closeness to issues of ESD may play an important role in teacher motivation and ultimately competent teachers, prior teacher education studies so far mostly neglected these components. This is problematic, as a further investigation of emotional and motivational variables is needed to advance professional development programs [40].

In prior studies, researchers used a wide variety of theories to explain teaching motivation [64]. One way of doing so is to draw upon general socio-psychological theories, like the *theory of planned*

*behaviour* (TPB [65]), which have been widely adapted to explain general environmental behaviour and also to predict intention to teach specific topics [66,67]. As several studies showed additional variables as predictor of intentions, researchers broadened the TPB resulting in the *model of goal-directed behaviour* (MGB [68]). The MGB includes emotions as additional predictors for behavioural intentions and further distinguishes this dependent variable into desires and intentions [68].

The MGB has already been successfully adapted as a model to investigate pre-service teachers' desires to teach about the return of the wolf [69]. In this and other studies [70], attitudes, enjoyment and perceived behavioural control towards teaching emerged as predictive factors for the motivation to teach about the return of the wolf. We do not know, however, what variables are in turn conductive for these attitudes, enjoyment and perceived behavioural control towards teaching the issue of returning wolves. In the original TPB these were called background factors and possible variables have often been debated [71]. In order to shed light on this question, we decided to use these variables as dependent motivational variables in the current study and investigate if any of the presented variables of values, motivations or closeness to the issue may constitute predictors for these factors [71].

As the study aims to draw conclusions on initial teacher professional development, we focused on pre-service teachers and addressed the following general research question:

**RQ:** What contextual and personality variables are motivationally relevant for pre-service teachers' motivation towards teaching about the return of the wolf?

To answer this superordinate research question, we developed a theoretical framework including specific relationships between the personality and motivational variables based on findings from environmental and wildlife psychology [72].

## 2. Theoretical Framework

### 2.1. Teaching Motivation

*Attitudes towards teaching about the return of the wolf* as a first motivational variable in our study describe positive or negative evaluations regarding teaching the respective topic in schools [65]. These evaluations have an affective as well as cognitive character; they concentrate on specific objects and have been found as predictors of behaviour in earlier studies [73]. In prior ESD studies, such attitudes have been also found to be linked to pre-service teachers' learning processes in sustainability issues [74].

Another, purely affective motivational variable is the emotion of *enjoyment towards teaching about the return of the wolf.* Emotions can be differentiated to attitudes in terms of their duration, complexity and the object of affective reaction [75]. Individuals experience enjoyment in positively valued situations with high controllability [76] and the emotion was also found as the prevalent positive teaching emotion [77]. Besides the connection to teaching motivation [69], enjoyment also correlated with other internal as well as external educationally relevant variables, like job satisfaction or student ratings of positive teacher behaviour in earlier studies [78].

Finally, *perceived behavioural control towards teaching about the return of the wolf* describes the individuals' beliefs about the perceived control of their own ability to perform a specific behaviour and is therefore conceptually similar to self-efficacy [65]. While both variables refer to the individual's ability to perform behaviours, perceived behavioural control better captures the specific nature of teaching about topics, as self-efficacy is a more general trait than concentrated on one behaviour [79,80]. Prior studies found knowledge [81] and initial teacher education [82] as important antecedents of efficacy beliefs.

### 2.2. Contextual Variables

#### 2.2.1. Personality Traits

As a first group of contextual variables we included personality variables like the wildlife value orientation of mutualism, attitudes towards the wolf and the protection motivation towards the species

in our theoretical model. These variables have shown to motivate the protection of wildlife species [72] and may explain some of the relevant personality factors, which constitute additional background factors for the respective behaviour [71].

Protection motivations are defined as behaviours which harm the environment as little as possible or even protect its resources [83]. These behaviours may occur in different settings, such as public or non-public behaviours [84] and can be either concentrated on general or specific manifestations of environmental behaviour [67]. Therefore, the *protection motivation towards the wolf* represents a specific motivation for harming wild wolves as little as much or protecting the species in public and non-public occasions. The increase in the motivation to protect the species is one requirement for a higher acceptance of wolf recovery in many parts of the world [10,15].

As described above, attitudes are positive or negative evaluations of contexts and are predictive of intentions and subsequent behaviours [65]. In addition to attitudes towards teaching the specific issue, there are also the *attitudes towards the wolf*, which concentrate purely on the species and evaluate its existence [85]. In combination with other social and contextual factors, such attitudes contribute to fostering acceptance about the protection of the species [86] and protection motivations [10]. But while attitudes might lead to subsequent motivations, attitudes also originate in deeper factors like values [13]. As described above, one prevalent value type for the return of the wolf may be wildlife value orientations.

Generally, wildlife value orientations describe the fundamental values, which people assign to wildlife [87]. These wildlife values can be differentiated into a mutualistic and a dominant value dimension. While *mutualism* constitutes a positive and protectionist view of wildlife, individuals with a higher domination oriented value set place humans over wild animals and assign society the right to decide over wildlife and use their resources [88]. Within these two dimensions, different subcategories may be differentiated [87]. One example may be the subdimension of *caring beliefs*, which capture how strong individuals personally care for animals and their welfare [88]. As prior studies found mutualism as the basic foundation for protection motivation in the context of returning wolves in Germany [57], we integrated this variable in our theoretical model.

Generally, young people with better education hold more positive attitudes and a higher general protection motivation towards the wolf [58]. As our sample of pre-service teachers is characterized by a young age and good educational background, we propose the following research hypotheses for the investigation about the connections between the presented personality traits and the motivational variables:

**H1:** Protection motivation is positively connected to the attitudes, enjoyment, as well as the perceived behavioural control towards teaching about returning wolves.

**H2:** Attitudes towards the wolf are positively connected to the attitudes, enjoyment, as well as the perceived behavioural control towards teaching about returning wolves.

**H3:** The wildlife value dimension mutualism is positively connected to the attitudes, enjoyment, as well as the perceived behavioural control towards teaching about returning wolves.

### 2.2.2. Perceived Closeness to the Issue

As described above, the perceived or real closeness to the return of the wolf might be connected to the motivation to teach, as prior studies found effects on behaviourally relevant variables like attitudes towards wolves [58,89]. Different to other studies, which directly measured the distance to wolf territories [61], we decided to utilize a socio-psychological approach to perceived distance. It is possible that people might not feel concerned by the return based on their profession, even if they live nearer to wolves. Therefore, we selected the construct of *psychological distance* [90].

This variable has already been extensively applied to study the closeness to climate change [59] and comprises of four subdimensions of geographical, temporal, social and hypothetical distance [91].

In the context of returning wolves, a psychologically distant individual will perceive wolves to return not in close proximity to her or him (geographical), not in the near future (temporal), not to her or him personally (social) and be generally relatively unlikely (hypothetical). In prior studies a decrease of psychological distance lead to higher environmental motivations [92], we propose the following hypothesis concerning this variable:

**H₄:** Psychological distance is negatively connected towards the attitudes, enjoyment and perceived behavioural control towards teaching about returning wolves.

### 2.3. Order of Variables Based on Cognitive Hierarchy

As described above, the original framework of the TPB allows for the integration of further background factors, which in turn predict the behaviourally relevant variables [71]. Therefore, all of the presented variables can be positioned as antecedents of the subsequent MGB variables.

To further classify the order of the tested variables to each other, we utilize the *theory of cognitive hierarchy*, which describes deeper factors like values as the basis for higher-order attitudes and more nuanced behaviours [13]. While the theory originated in wildlife research [87], other researchers found similar structures within domains like environmental psychology. One example may be the value-belief-norm theory [84], which also has been applied to study the pro-environmental behaviour of students [93]. While the terminology may be different, the structure of deeper and more general variables, which then in turn predict more specific variables, is similar.

While the deeper factors are more general and also slow to change, the higher order variables are more specific and also faster to change. Based on this assumption, the variable of mutualism should be predictive of the attitudes towards the wolf, which will in turn be predictive of the protection motivation towards the species. While protection motivation should then be directly connected to the motivational variables, psychological distance might be directly connected to the motivational variables. Therefore, we state our final hypothesis:

**H₅:** The variables are ordered in accordance to the theory of cognitive hierarchy.

A graphical overview of the theoretical framework of our study is shown in Figure 1.

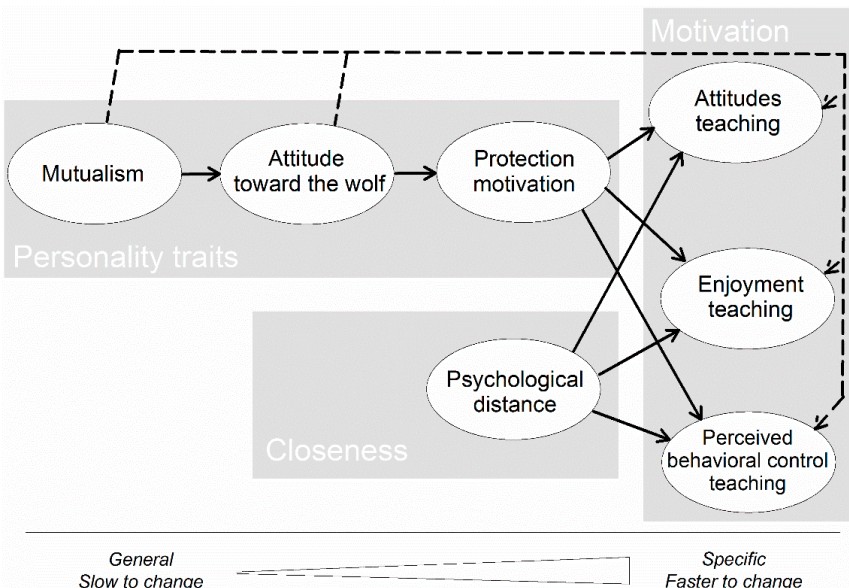

**Figure 1.** The theoretical model of the study testing for connections between contextual variables of personality traits and closeness to the issue of returning wolves with the motivational factors to teach about returning wolves according to the theory of cognitive hierarchy. Bold lines indicate direct and dashed lines indirect effects.

## 3. Materials and Methods

### 3.1. Research Design and Sample

To investigate the superordinate research question and address the stated hypotheses, we followed a cross-sectional research design and created a self-completion questionnaire [94]. The questionnaire was distributed in a German paper-and-pencil version in several lectures at a university in Lower Saxony, a federal state in the northwest of Germany. Because no special randomization was implemented, the sample represents a convenience sample [94]. Prior to the study, several sightings of wolves were reported in the surroundings of the city and in Lower Saxony in general [8,95].

The analysed sample consisted of 120 pre-service biology teachers, 98 of them female (82.4%) and 21 male (17.6%). One participant did not report the corresponding gender but was not excluded from the analysis as gender was not investigated further in the current study. Even though the number of female participants is greater than the number of male participants, the sample appears to reflect the gender distribution in the desired population of pre-service biology teachers, as other studies had similar sample structures [96]. Participants ranged in age from 19 to 38 years (M = 23.2 years, *SD* = 3.3 years).

The study was conducted in accordance with the Declaration of Helsinki, the German Research Foundation (DFG) as well as the American Psychological Association (APA) and we obtained informed consent for inclusion in the sample from every participant before the study. Furthermore, we ensured anonymity of all participants, who also had at any moment the possibility to skip single questions or the whole questionnaire. The protocol was therefore not approved by a local ethics committee, as the research had no medical background, assessed no sensitive personal information and all participants were introduced to the aim of the study.

### 3.2. Questionnaire

The scales relevant for the current study were integrated in a greater questionnaire about the teaching motivation of pre-service teachers, which are published elsewhere [69]. All variables except of gender were measured using a 6-point Likert scale. The original questionnaire was distributed in German and the corresponding items were translated into English for the purpose of this paper. The translation of the original scales into German was achieved using a double-translation approach by back translating them from German to English and checking for coherence. The wording of all items can be obtained from the Appendix A.

#### 3.2.1. Contextual Variables

To measure the mutualistic dimension of wildlife value orientations, we used the established original wildlife value scales [88]. We decided to measure the mutualistic subdimension of caring beliefs, as these showed to be the most relevant based on prior studies [97]. The scale overall compromised four items.

To measure attitudes towards the wolf, we selected the general attitude towards wildlife measure of Kaczensky, Blazic & Gossow [85] and adapted it to the chosen animal context. This was done by replacing the word "bear" in the original items with the word "wolf." Furthermore, we changed the original destination in one item (Slovenia) to Germany, as this was the selected country where the study was done. While the overall scale comprised five items, the final item was reverse coded similar to the original scale.

The scale for the protection motivation towards the wolf was constructed using the items of Stern et al. [98]. We closely adapted these items to the context of the protection of the wolf, similarly to the attitudes towards the wolf. This resulted in a three item scale, which included private-sphere behaviours as well as public behaviours to measure protection motivation coherently to our theoretical framework [84].

The scale for the measurement of psychological distance was constructed based on the four subdimensions of the variable [91], which were each measured with one item. This is similar to the adaption of the scale in prior studies [60]. This means that each participant rated his or her concern about the return of the wolf in terms of the social, temporal, geographical and hypothetical manifestation of the phenomenon. After the data collection, all items of this scale were reverse coded, since the overall scale is negatively defined.

### 3.2.2. Motivational Variables

The scales of the motivational variables were constructed using established scales of the MGB [99]. Similar to the other scales we aimed for a direct adaptation by only replacing the respective objects of the items. Different to the original MGB we measured emotions using discrete emotions, instead of the dimensional emotions from the original MGB [100]. The emotion of enjoyment towards teaching was measured using the Differential Emotions Scale [101]. The stimulus was presented by an introductory sentence, which was "If I imagine teaching the topic 'the return of wolves' in biology class, I feel ...." Participants were then asked to rate their agreement to experiencing the three respective adjectives. Finally, we addressed PBC by asking the pre-service teachers for their subjectively perceived possibilities of teaching the chosen context.

### 3.3. Statistical Analysis

### 3.3.1. General Approach

As a first step, we performed a confirmatory factor analysis (CFA) to ensure the discriminant validity of all scales [102]. After inspecting Cronbach´s alpha as a further measure of internal consistency [103], we investigated relations between the variables based on their correlations (Table 1). Following our theoretical assumptions, we then calculated a structural equation model (SEM) and tested the predictive ability of the variables ($H_{1-4}$) with the hypothesized structure, building on the theory of cognitive hierarchy ($H_5$).

As an additional test of this structure, we calculated an additional structural equation model with direct effects of the deeper variables on the motivational factors and compared it to our theoretical model. In this additional model, only the most specific contextual factors of psychological distance and protection motivation were predictive of the attitudes, enjoyment and perceived behavioural control towards teaching. This additionally indicated the direct effects of these variables and indirect effects of mutualism and attitudes towards the wolf according to the selected theoretical model.

To further illustrate the differences for pre-service teachers' motivational factors based on their psychological distance and protection motivation, we finally split the sample into two groups according to a high or low scoring on these scales ("median split") and compared these groups according to the attitudes, enjoyment and perceived behavioural control towards teaching about the return of the wolf [104].

### 3.3.2. Applied Procedures

Due to the skewness and kurtosis of several variables (see Table 1), we selected robust statistical procedures. For the calculations of all models we therefore used a robust maximum-likelihood estimator (MLR), which showed to be robust against the violation of several assumptions like non-normality [105]. We also used robust methods for the correlations (correlation coefficient spearman-rho) and difference tests (Mann-Whitney-U for group comparisons).

As recommended by Kline [106], we evaluated the model fit by combining the fit indices of the root mean square error of approximation (RMSEA), Bentler comparative fit index (CFI) and the standardized root mean square residual (SRMR). Sufficient model fit is indicated by a RMSEA less than or equal to 0.08, a CFI greater than or equal to 0.95 and a SRMR less than or equal to 0.08 [107]. Possible model modifications were based on the combination of model modification indices in accordance with

theoretical considerations [102,106]. To test for indirect effects of the variables, we performed a Sobel test [105].

While the CFA and SEM were performed using Mplus 7.3 [105], we used IBM SPSS 24 for the descriptive statistics, the calculation of Cronbach's alpha, as well as the correlations and difference tests between the variables. Results from the Mann-Whitney-U test were converted to a standardized effect sizes for better comprehension. All presented values represent standardized values.

### 3.3.3. Measurement Results

The first estimation of the data led to an unacceptable fit of the CFA model, with some values fitting but others not passing the selected criteria ($\chi^2$ = 460.906 (231, 0.00), RMSEA = 0.09, CFI = 0.88, SRMR = 0.09). Further inspection indicated a measurement problem with one item from the psychological distance scale and intercorrelations between several items from similar scales (e.g., perceived behavioural control and wildlife values). We therefore decided to modify the measurement model based on empirical as well as theoretical justifications [102,106].

As a first step, we excluded the last item (hypothetical distance; PD04) from the psychological distance scale. We think the unacceptable loading is due to the non-plausibility of the item in the selected sample. The wolf already started to establish near the study site [8], so the participants may have seen no sense on indicating how likely the return will occur. The psychological distance scale still encodes the spatial, temporal and social distance to the process, so this exclusion does not impede the overall measurement of closeness to returning wolves. Besides this, we finally added five intercorrelations to the model (see the Supplemental Material for further information). We decided to add these correlations as a better way to address the problem than deleting single items, which is most often theoretically unjustified [102].

The estimation of this modified model led to an acceptable fit for the CFA [$\chi^2$ = 314.559 (225, 0.00), RMSEA = 0.06, CFI = 0.95, SRMR = 0.06] and overall good factor loadings. The very good values for Cronbach's alpha (Table 1) also indicated a good internal consistency. Therefore, we accepted the model under the given modifications and continued with the further analysis.

## 4. Results

### 4.1. Descriptive Results and Correlations

As described in Table 1, most of the variables showed a slightly negatively skewed distribution. This concerned almost all variables except for mutualism and attitudes towards the wolf. The skewness was mainly due to relatively high values for the motivational variables of attitudes (*M* = 4.38; *SD* = 0.90; *Mdn* = 4.33), enjoyment (*M* = 3.91; *SD* = 0.94; *Mdn* = 4.00) and perceived behavioural control (*M* = 3.86; *SD* = 0.98; *Mdn* = 4.00) towards teaching. The skew also concerned protection motivation (*M* = 3.66; *SD* = 1.04; *Mdn* = 3.66). The highest value was reported for psychological distance (*M* = 4.79; *SD* = 1.14; *Mdn* = 5.00).

Concerning the connections between the variables, almost all of the contextual personality traits correlated with the attitudes, enjoyment and perceived behavioural control towards teaching (Table 1). While almost all connections had a medium effect size, especially protection motivation was positively correlated to the attitudes (*r* = 0.52, *p* < 0.001) and enjoyment (*r* = 0.65, *p* < 0.001) with a large, as well as to perceived behavioural control towards teaching with a medium effect size (*r* = 0.46, *p* < 0.001). The strongest relations were found between the dependent variables, which all showed positive correlations with large effect sizes (*r* = 0.48–0.63, *p* < 0.001).

**Table 1.** Bivariate Spearman-rho correlations and descriptive statistics of the variables.

| Variable | 1 | 2 | 3 | 4 | 5 | 6 | 7 |
|---|---|---|---|---|---|---|---|
| 1. Mutualism | - | | | | | | |
| 2. Attitudes wolf | 0.34 *** | - | | | | | |
| 3. Protection motivation | 0.43 *** | 0.59 *** | - | | | | |
| 4. Psychological distance | −0.22 * | −0.04 | −0.01 | - | | | |
| 5. Attitudes teaching | 0.24 ** | 0.33 *** | 0.52 *** | −0.27 ** | - | | |
| 6. Enjoyment teaching | 0.49 *** | 0.46 *** | 0.65 *** | −0.15 | 0.63 *** | - | |
| 7. PBC teaching | 0.31 ** | 0.19 | 0.46 *** | −0.23 * | 0.48 *** | 0.61 *** | - |
| Items | 4 | 5 | 3 | 3 | 3 | 3 | 3 |
| Mean | 3.50 | 4.39 | 3.66 | 4.79 | 4.38 | 3.91 | 3.86 |
| Median | 3.50 | 4.40 | 3.66 | 5.00 | 4.33 | 4.00 | 4.00 |
| Standard deviation | 1.05 | 0.82 | 1.04 | 1.14 | 0.90 | 0.94 | 0.98 |
| Skewness | −0.05 | −0.41 | −0.71 ** | −0.88 ** | −0.52 * | −0.61 ** | −0.68 ** |
| Kurtosis | 0.00 | 0.49 | 0.60 | 0.15 | 1.38 ** | 1.21 ** | 1.08 * |
| Cronbach's $\alpha$ | 0.87 | 0.89 | 0.85 | 0.86 | 0.86 | 0.92 | 0.84 |

Note. * = $p < 0.05$, ** = $p < 0.01$, *** = $p < 0.001$. PBC = perceived behavioural control. Significant skewness and kurtosis indicate a significant deviation from normality [103].

### 4.2. Structural Equation Model

The results of the structural equation model are depicted in Figure 2. Mutualism positively predicted the attitudes towards the return of the wolf ($\beta = 0.40$, $p < 0.001$), which in turn positively predicted the protection motivation of the wolf ($\beta = 0.63$, $p < 0.001$). This protection motivation was a very strong direct predictor of the attitudes ($\beta = 0.71$, $p < 0.001$), enjoyment ($\beta = 0.68$, $p < 0.001$), as well as PBC towards teaching ($\beta = 0.52$, $p < 0.001$). Psychological distance was another but overall weaker negative direct predictor of the attitudes ($\beta = −0.39$, $p < 0.001$), enjoyment ($\beta = −0.22$, $p < 0.01$), and PBC towards teaching ($\beta = −0.31$, $p < 0.01$).

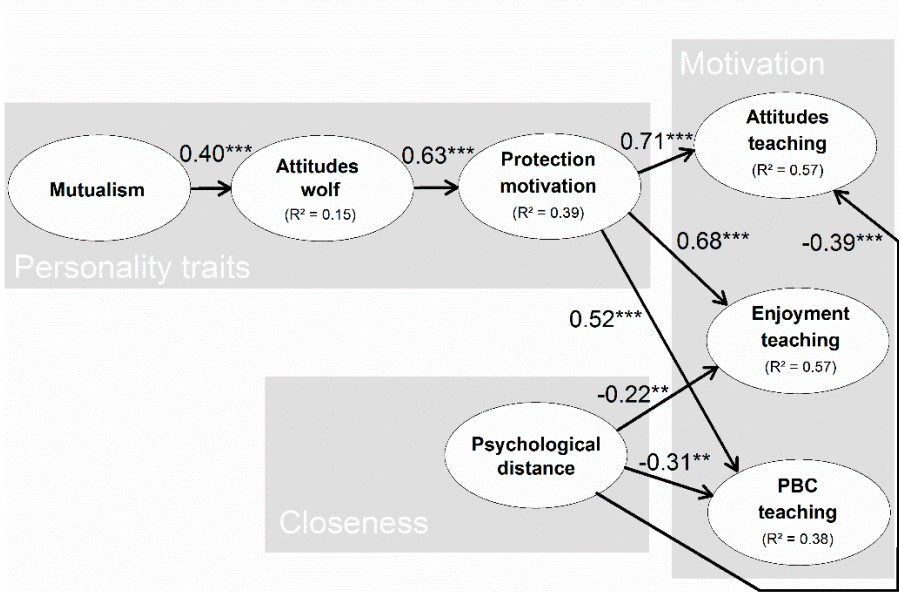

**Figure 2.** Structural equation model for the attitudes, enjoyment and perceived behavioural control (PBC) towards teaching. Indirect effects and loadings are omitted for clarity. Indirect effects are displayed in Table 2 and the loadings can be obtained in the supplemental material. Model fit: RMSEA = 0.06, CFI = 0.95, SRMR = 0.08. ** = $p < 0.01$ *** = $p < 0.001$, $R^2$ = explained variance of the latent variable.

Overall, attitudes towards the wolf showed with 15% the lowest share of explained variance ($R^2 = 0.15$). For the variables of protection motivation ($R^2 = 0.39$) and perceived behavioural control

($R^2 = 0.38$) the predictors explained overall more variance, with 38% and 39% of the explained variance. Finally, the predictors of attitudes ($R^2 = 0.57$) as well as enjoyment towards teaching ($R^2 = 0.57$) explained the variance in these variables to a great extent, as the predictors explained more than half of the variance in both variables.

The indirect effects of mutualism and attitudes towards the wolf are further displayed in Table 2. Both variables showed no direct but indirect effects through protection motivation. Overall, attitudes towards the wolf predicted a substantial amount of the attitudes ($\beta = 0.45$, $p < 0.001$), enjoyment ($\beta = 0.43$, $p < 0.001$) and PBC towards teaching ($\beta = 0.33$, $p < 0.01$). Mutualism also showed indirect effects, which were overall weaker.

With regard to the total effects, protection motivation was clearly the strongest predictor for all of the three dependent variables. Attitudes towards the wolf still showed remarkable effects, which were slightly greater than the relations between psychological distance and the dependent variables. Mutualism only predicted variance for the enjoyment towards teaching ($\beta = 0.29$, $p < 0.01$).

**Table 2.** Direct, indirect and total standardized regression effects ($\beta$) of the predictors for the dependent variables of attitudes, enjoyment and perceived behavioural control towards teaching with corresponding significance levels ($p$).

| | Attitudes Teaching | Enjoyment Teaching | PBC Teaching |
|---|---|---|---|
| **Direct effects** | | | |
| Mutualism | −0.16 | 0.14 | 0.00 |
| Attitudes towards wolf | −0.09 | −0.05 | 0.00 |
| Protection motivation | 0.71 *** | 0.68 *** | 0.52 ** |
| Psychological distance | −0.39 *** | −0.22 ** | −0.31 ** |
| **Indirect effects** | | | |
| Mutualism$^{ATTWOLF+PROT}$ | 0.14 * | 0.15 ** | 0.13 * |
| Attitudes towards wolf$^{PROT}$ | 0.45 *** | 0.43 *** | 0.33 ** |
| Protection motivation | - | - | - |
| Psychological distance | - | - | - |
| **Total effects (direct + indirect)** | | | |
| Mutualism | 0.02 | 0.29 ** | 0.13 |
| Attitudes towards wolf | 0.36 ** | 0.38 *** | 0.33 ** |
| Protection motivation | 0.71 *** | 0.68 *** | 0.52 ** |
| Psychological distance | −0.39 *** | −0.22 ** | −0.31 ** |
| Overall explained variance ($R^2$) | 0.57 | 0.57 | 0.38 |

Note. * = $p < 0.05$, ** = $p < 0.01$, *** = $p < 0.001$. PBC = perceived behavioural control, $^{ATTWOLF+PROT}$ = through attitudes towards the wolf and protection motivation, $^{PROT}$ = through protection motivation.

### 4.3. Group Comparisons for Psychological Distance and Protection Motivation

As psychological distance and protection motivation emerged as only direct predictors, we used a median split and compared high and low scoring individuals on these scales concerning their attitudes, enjoyment and perceived behavioural control towards teaching. As the results illustrate, we found significant differences between the high and low scoring individuals for almost all dependent variables (Figures 3 and 4).

First of all, we found significant differences between higher for the protection of the wolf motivated (score of 4 and above) and less for the protection motivated pre-service teachers (score under 4) for the attitudes ($Z = -5.160$, $d = 1.074$, $p < 0.001$) enjoyment ($Z = -6.205$, $d = 1.383$, $p < 0.001$) and also for the perceived behavioural control towards teaching ($Z = -4.22$, $d = 0.889$, $p < 0.001$).

For the split of the psychological distance, we found significant differences between psychologically distant (score above 5) and psychologically more close individuals (score 5 or smaller) for the attitudes ($Z = -3.068$, $d = 0.586$, $p < 0.01$) and perceived behavioural control ($Z = -2.114$, $d = 0.414$, $p < 0.05$) but not for enjoyment towards teaching ($Z = -1.175$, $d = 0.217$, $p > 0.05$).

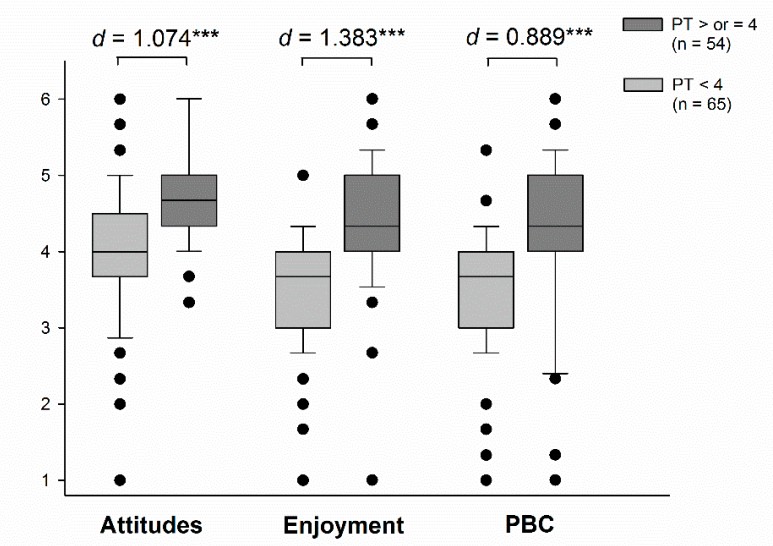

**Figure 3.** Group comparisons for the highly (value of 4 and above, dark grey) and small protection motivated individuals (value smaller than 4, light grey) for the variables of attitudes, enjoyment and perceived behavioural control (PBC) towards teaching about the return of the wolf with results from Mann-Whitney-U-test (Cohens *d* with corresponding *p*). *** = $p < 0.001$.

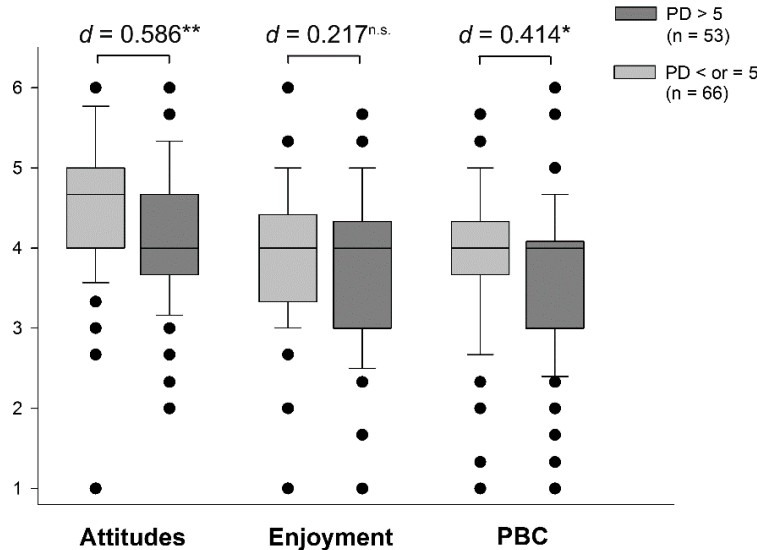

**Figure 4.** Group comparisons for the psychologically distant (value above 5, dark grey) and psychologically more close individuals (value 5 or smaller, light grey) for the variables of attitudes, enjoyment and perceived behavioural control (PBC) towards teaching about the return of the wolf with results from Mann-Whitney-U-test (Cohens *d* with corresponding *p*). n.s. = $p > 0.05$, * = $p < 0.05$, ** = $p < 0.01$.

## 5. Discussion

### 5.1. Relations among Contextual and Motivational Variables

#### 5.1.1. Protection Motivation towards the Wolf (H_1)

The strong relation between protection motivation and the three motivational variables of attitudes, enjoyment and perceived behavioural control towards teaching about returning wolves shows how a protection motivation about a species translates into higher motivation for teaching about the respective species. This result supports our hypothesis, as the positive direct prediction as well as the group comparisons indicated the hypothesized positive connection between the variables.

This is also the first result with a positive link between a contextual motivation to protect a species and a specific teaching relevant variable. Such personal contextual factors have been suggested as influences of teacher behaviour in ESD [32] but have only scarcely been investigated in prior studies. Moreover, our findings close a gap in prior research, as other studies either concentrated on teacher's protection motivation towards species or on teaching about the species but only few studies investigated the relation between these two perspectives. Our results also underline normative statements, which emphasize the motivation of pre-service teachers' sustainable behaviours as an aim of teacher education [108].

Transferred to other contexts, this connection implies a positive relationship also for other topics of ESD, for example climate change, energy saving or waste reduction. For these topics, the motivation to protect the climate, save energy or reduce waste should also translate to a higher motivation to teach about these topics. Nonetheless, it may be debatable, how specific these protection motivations should be assessed. This is also an open discussion in environmental psychology. Prior studies showed robust connections between different specific protection motivations like energy conservation, recycling or sustainable transportation [109]. These connections would implicate one general protection behaviour as sufficient, due to the same underlying motivations. In this case, a general approach to teacher professional development may be sufficient. But in empirical investigations specific predictors outperformed general predictors for their predictive ability for environmental behaviours [67]. Transferred to connections between nature protection and teaching, specific protection motivations should also outperform general variables for their predictive ability.

This is at the moment only a hypothesis but it might be important to further investigate this to assess the possibility of using protection motivation as a possible target of interventions and explicit instruction. As our results show, this might be a good way of increasing teaching motivation. But nonetheless, such an intended increase of specific protection motivations in teacher education might be challenging, since a wide variety of deeper personality variables determines protection motivation [110]. As our study showed, deeper personality traits of attitudes towards the wolf and wildlife values build the foundation of pre-service teacher's protection motivation.

### 5.1.2. Attitudes towards the Wolf (H$_2$)

Concerning the attitudes towards the wolf, we found this deeper personality factor as an indirect predictor of the attitudes, enjoyment and perceived behavioural control towards teaching about the return of the wolf. Overall, attitudes towards the wolf showed a smaller connection to the dependent motivational factors than the protection motivation. But according to our hypothesis, attitudes towards the wolf were highly predictive of the protection motivation, which is why they should nonetheless be reflected in teacher professional development in ESD.

These results are in line with prior studies, which showed a positive connection between environmental attitudes and environmental behaviour [111]. Beliefs and attitudes are one integral part of teacher identity and might promote or hinder pre-service teachers in integrating ESD contexts into teaching, especially if personal beliefs and attitudes are not in harmony with ESD-related values [112]. Consistent to this, attitudes towards the wolf served as the foundation of the protection motivation towards the wolf in our study. The connection with the motivational variables is also in line with studies from other domains, which successfully correlated context-specific attitudes with the preference to teach about the respective context. Examples stem from the domains of teaching about the evolution of species [113] and climate change [33,114]. Similar to these studies, the participants of the recent study also reported slightly skewed distributions of several variables. This especially concerns the variable of attitudes towards the wolf. While the selected sample of biology teachers may indeed hold more positive attitudes towards animal protection than other sample populations given their selection of biology as a subject, future studies could try to integrate other data types than self-report to triangulate the findings from these self-report measures.

Despite these design-based limitations, the results are important for teacher professional development, as differences in attitudes may lead to differences in the success of professional development activities [40]. Therefore, pre-service teachers with more positive attitudes towards the wolf may acquire knowledge in teacher education easier as other pre-service teachers, which was also implied by a study about attitudes and knowledge about climate change [114]. While this should be investigated in further studies, attitudes towards the respective animals should obviously be addressed in teacher education.

This concerns, for example, the aim of higher education to foster future teachers' positive attitudes towards teaching about sustainability-related issues [115]. Due to their position in the cognitive hierarchy, attitudes might be easier to change than values. Values are of a more general nature and therefore more difficult to change once they have been developed [13,116]. Changing attitudes might nonetheless be difficult for the case of returning wolves, as direct experiences are not possible or could lead to declining attitudes [89]. Therefore, possible interventions should focus on either positive examples of human-wolf-coexistence [117] or use a system approach and give new information about how the ecological, economic and social dimensions of the conflict interact with each other. Besides these opportunities to increase attitudes, our results imply wildlife values as the deepest foundation of the protection motivation and attitudes towards the wolf [13]. Therefore, attitude change might be not possible without changing the factors causing the attitudes.

### 5.1.3. Wildlife Value Orientations (H₃)

As described in the results section, the wildlife values of caring beliefs were an indirect predictor of the enjoyment to teach about the return of the wolves but neither a predictor of the attitudes nor the perceived behavioural control towards teaching. While this result fits to the direction of our hypothesis, the connections were not as strong as suggested by our theoretical model, as the wildlife values only predicted one of the motivational variables.

We believe this result is explainable by the nature of emotion causation. While there is still discussion about this causal nature in general [118], all theories define that emotions describe reaction to a specific stimulus, which then causes the individual to react, based on prior experiences [75,119]. Qualitative results about teachers' emotions showed that individual identity and biography are important factors for teachers' emotional reaction on a particular stimulus [120]. The wildlife values of our study may finally be a measure of this identity and biography, as values are deeply connected to one's own life and experiences [121].

While this is in line with general emotion theory [122], prior research on teacher emotions has regarded reactions to student actions as a main cause of teacher emotions in general [77]. Our results indicate for the first time a different and more topic-specific cause of teacher emotions. While it is clear that teachers might react in specific situations involving student misbehaviour with negative emotions [123], there might still be differences in the teachers' appraisals when they teach a positively valued topic compared to a negatively valued topic. For example, individuals with a higher mutualistic view of the world may show more resilience against disruptive student behaviour in lessons about returning wolves. This should be further investigated in a comparative study with in-service teachers.

Whereas these results clarify the causation of teaching emotions, the investigation of values as part of teachers identity points to problems based on an ethical dilemma of teacher professional development. When we know the coherent values for specific aims of education, should teachers be purposefully selected to fit to these values? Or should we try to change these values in higher education? This is especially problematic as societal values might change. First of all, our study explicitly not aimed at investigating pre-service teacher personality in means of excluding inappropriate candidates but rather analysing successful candidates to foster learning in higher education. Furthermore, teacher professional development about issues like returning wolves should not force learners to adopt any desired way of thinking but build students capacity to manage their own life in a changing environment [124]. Vare and Scott [124] described this capacity building as ESD

2, which also includes contradictions and therefore is different to ESD 1 (giving clear information and motivating in the short term).

Therefore, in professional development activities, students should reflect about their own values about topics like returning wolves. In this view, the role of teacher educators might shift to enable students' reflection processes by providing the right environment and suitable learning materials and topics. This directly points to the next hypothesis, as topic selection is at the core of teacher educators' work and psychological distance was the final predictor of the motivational variables.

### 5.1.4. Psychological Distance ($H_4$)

Besides the personality traits towards the species, psychological distance was the second direct predictor of pre-service teachers' attitudes, enjoyment and PBC towards teaching about the return of the wolf. In our model, psychological distance was negatively connected to the motivational variables. Therefore, pre-service teachers with less psychological distance to the return of the wolves showed more positive attitudes, enjoyment and perceived behavioural control towards teaching about the issue. This was also illustrated by the group comparisons in which those who scored higher in psychological distance were compared to those with a lower score. According to the results from the regression we found significant differences between the two groups for the attitudes and perceived behavioural control (lower in the high-score group for psychological distance). Only for the enjoyment towards teaching there was no difference between the groups.

This mostly supports our initial hypothesis and seems logical in terms of prior research about relevant processes in educational research, as an increased relevance based on a more close feeling to the process of returning wolves should contribute to a higher motivation to learn about the topic [62]. But as this effect was not yet tested in a pre-service teacher sample, we now have a more specific indicator and a further possible antecedent of teaching motivation.

For the concrete design of teacher preparation courses, this implies the selection of topics with a small psychological distance within these courses, as the increased closeness to the issue might facilitate the learning of content or pedagogical content knowledge based on a higher interest [125]. As the scale of psychological distance is rather new to educational research, a further test of the applicability to teacher training is needed. Furthermore, the results of the present study should be only transferred cautiously, as the surveyed pre-service teachers generally reported a high psychological distance towards the return of the wolves (mean of 4.79 and a median of 5). While this might be explained by the teaching profession being far away from the problems with returning wolves, future studies should comparatively look in how this psychological distance might differ between specific teaching contexts.

Another interesting finding was the missing connection between psychological distance and the protection motivation or the attitudes towards returning wolves. Within the topic of returning wolves, there were already studies who found more negative attitudes and less protection motivation for individuals living closer to or being affected by wolves [57,61]. While prior experimental studies showed how a decreasing psychological distance might lead to a higher engagement with climate change [92], our results show a more nuanced view, as there may be differences in how individuals experience psychological distance based on the context and their belonging to different groups [59]. Future studies could further investigate in how teachers might differ to stakeholders of returning wolves or the general public concerning their evaluation of psychological distance to specific processes.

### 5.2. Structure of Variables ($H_5$)

As shown by the direct and indirect effects, we found a hierarchical structure of the variables, which was implied by prior literature and also corresponds with our hypothesis. Due to this hierarchical structure, wildlife values can be regarded as the foundation of positive attitudes towards the wolf, which in turn foster the protection motivation towards the species. Finally, this protection motivation was a significant contributor to the motivation to teach about the wolf.

In terms of teaching and learning about of the return of the wolf, this might be a difficult message, since values are deep factors of personality, which are difficult to change [13,116]. This view generally complicates societal value change for nature protection [38]. While such a change may be too ambitious in the context of teacher education, changes in individual values might possibly lead to greater changes in systems. Similar research described such changes as "leverage points" [126]. The selection of a deep leverage, like changing goals or values deep in a system might therefore lead to greater positive changes in the whole system [126]. This underlines the normative character of education in addressing learners not only on cognitive levels but also on social-emotional and behavioural levels [19].

Nonetheless, this change might be impossible due to the difficult nature of values, the high influential potential of the closer personal social surrounding such as parents and family and the diverse interests in societies [38]. Furthermore, local beliefs and cultures might play a difficult role, as studies have shown differences in wildlife values based on (local) culture [127]. Yet, teacher education should not give up on this issue. Making teachers aware of the role of values in a person's general view on ESD-related issues may shape their view on educational interventions that address children's value perspectives. While future teachers are still in their higher education system, they may also want to learn about relevant teaching strategies. Therefore, universities should take the responsibility to offer such holistic learning opportunities. Examples are classes in which students reflect on their own value basis while at the same time learning about the theoretical assumptions surrounding the role of values for facing the challenges of a sustainable future. This could also include the economic dimensions, illustrating the ecosystem services provided by wolves.

### 5.3. Affective and Ultimately Transformative Teacher Professional Development in ESD

As described by our results and discussed before, we found values, attitudes and psychological distance as important contributors to pre-service teachers motivation to teach about returning wolves. This is an important new insight for teacher professional development, as motivation is described as one part of competent teachers in professionalization models like the action competence model [47]. When we normatively expect teachers to be motivated when teaching about topics of ESD, then the presented personality values may be of importance and need to be reflected in professional development activities.

While our study not aimed for the investigation of specific methods for successful professional development activities, the results of our study suggest higher teaching motivation in participants when teacher educators address affective learning outcomes like attitudes and values in accordance with prior literature [40]. Like these prior studies showed, this could be realized by integrating deeper learning experiences instead of only acquiring new knowledge [128]. Concrete examples could include the integration of external stakeholders and real-world experiences in relevant issues in teacher professional development [129–131]. These real-world experiences might connect the cognitive input with interdisciplinary characteristics of the issues and needed affective appeals [132]. For the issue at hand, this would ideally mean to invite stakeholders into the lessons. But as this might not always possible in formal education, educators should try for the most authentic learning experience. These could for example also be assisted by media contents. In addition, students and especially in-service teachers could be brought together in learning communities with a specific scope [133]. Such communities could provide new knowledge, teaching materials or social connections, which are further requirements for sufficient motivation to teach ESD issues [134]. Similar learning communities in institutions of higher education might also foster the integration of such topics into higher education [135], which might be a main catalyst for curricular transformation [136].

While learners should always reflect on their experiences and value structure none of these affective experiences are by any means a silver bullet to foster deeper personality change in pre-service and in-service teachers. Like the effect of psychological distance in our study showed, motivation and also learning are contextually bound processes [32]. Therefore, teacher professional development should reflect this and look for specificity instead of general recipes in teacher education. This directly

points to the start of the discussion, where we discussed the specificity of protection motivations. Of course it is not possible to tackle every ecological problem solely in ESD. Teacher educators should instead look for specific experiences, which probably also lead to transformational learning processes [53]. As described above, through such a learning process deeper personality factors like core beliefs and values might be better changeable and lead to a transformation of several overlaying behaviours, which also would be a "psychological" leverage point [126]. Therefore, future studies should investigate how these affective experiences can be integrated into approaches of transformative learning and also assess their long-term impacts on the individual and probably also groups. But based on our data, such transformations in higher ESD seem to be successful through changing deeper personal variables than classical intervention studies.

## 6. Conclusions

Our study showed how specific contextual factors about an exemplary ESD context are connected to the attitudes, enjoyment and perceived behavioural control of pre-service teachers to teach about the issue of returning wolves. Besides identifying specific variables which may be aim of interventions to foster the teaching motivation for the selected context, our study identified further, non-cognitive dimensions that should be considered in teacher professional development. This includes protection motivation, psychological distance but also deeper variables like values. For future studies it could be interesting to integrate other data types than self-report measures, as this could further rule out possible effects of biased answers. This could for example include physiological measures of emotional reactions like heart rate or skin conductance [137]. Nonetheless, results from self-report measures like the present study are needed to sufficiently design and explain such more objective approaches.

Finally, our study therefore stimulates new research in the domain of non-cognitive factors in teacher professional development in the context of ESD. As the belief system of teachers is likely to change when they enter school practice [138], our results are at the moment only applicable to pre-service teachers. But since teachers' individual experiences during teacher education are an important contributor to professional development [82], this phase of teacher education is important for shaping the professional identity of teachers. This has also been shown in biodiversity contexts [139]. Sound teacher education concepts may arm teachers for the challenges of school practice. For example, if they learnt how to reflect on their own value structure, they may become more capable in asking themselves of their teaching practice is still congruent with what they value. Sometimes, such reflective practices fall short under the pressure of challenging school routines. In higher education, we may be able to train future teachers in value reflection and try to transform them to more sustainable ways of living and teaching [140]. Of course these changes also require academic staff development in universities, which could for example be initiated by curricular changes towards ESD, as a qualitative study showed [141]. Such a curricular change might explicitly include ethics and the reflection of own values [142].

This transformation may not stop in general higher education but it should be explicitly integrated in teacher education [143]. Prior studies have shown differences between specific cultures and countries concerning the preparation of teachers in the domain of sustainability [144]. This also concerns specific subjects, which may differ in their integration of ESD, based on the different subject-related cultures [145]. Therefore, teacher educators should not rely on general solutions for all countries and subjects but should deliver subject-specific formats of teacher professional development. This is also underlined by the effect of psychological distance as a specific contributor to the motivation to teach about issues in ESD.

Overall, our study therefore allowed a first glimpse into how considering context-specific protection motivations, values, positive attitudes and the selection of contexts with a low psychological distance might translate into an increase in the teaching motivation of pre-service teachers in ESD. This increased motivation of pre-service teachers might be one stepping stone to strengthen ESD in schools and lead to a more sustainable future for everyone. This hopefully also leads to sufficiently

prepared future citizens to overcome human-wildlife conflicts and viable populations of wildlife animals like wolves.

**Supplementary Materials:** The following are available online at http://www.mdpi.com/2071-1050/11/1/47/s1, Table S1: Original SPSS data sheet (DancewithwolvesSPSSdata.sav), Table S2: Original Mplus data sheet (DancewithwolvesMplusdata.dat), File S3: Output file from CFA (CFA.out), File S4: Output file from SEM (SEM.out).

**Author Contributions:** Conceptualization, A.G.B.; Formal analysis, A.G.B.; Investigation, M.S.; Writing-original draft, A.G.B.; Writing-review & editing, A.G.B. and S.M.

**Funding:** This research received no external funding.

**Acknowledgments:** We thank all participating pre-service teachers. Parts of this article (in a manuscript version) are part of the publication-based dissertation of the corresponding author. We acknowledge support by Deutsche Forschungsgemeinschaft (DFG) and Open Access Publishing Fund of Osnabrück University.

**Conflicts of Interest:** The authors declare no conflict of interest.

## Appendix A

**Table A1.** Scales of the study and their corresponding items with results from confirmatory factor analysis (CFA).

| Construct | Item |
|---|---|
| **Mutualism (MUT)** | |
| MUT01 | I care about animals as much as I do other people. |
| MUT02 | It would be more rewarding to me to help animals rather than people. |
| MUT03 | I take great comfort in the relationships I have with animals. |
| MUT04 | I feel a strong emotional bond with animals. |
| **Attitudes towards the wolf (ATTWO)** | |
| ATTWO01 | I have a positive attitude towards the return of the wolves to Germany. |
| ATTWO02 | It is important for Germany to have a viable population of wolves. |
| ATTWO03 | Wolves living in Germany are important, even if I never see one. |
| ATTWO04 | Wolves are a sign of an intact nature. |
| ATTWO05 * | Because many wolves live in other countries, there is no need to have wolves in Germany, too. |
| **Protection motivation (PROT)** | |
| PROT01 | I would be willing to spend money to actively promote the return of the wolves. |
| PROT02 | I would support the recovery of the wolves at an election. |
| PROT03 | I would immediately sign a petition for the recovery of wolves. |
| **Psychological distance (PD)** | |
| PD01 * | I am personally concerned by the return of the wolves. |
| PD02 * | I am concerned by the return of the wolves in my geographical surroundings. |
| PD03 * | I am concerned by the return of the wolves in the near future. |
| PD04 * | The return of the wolves is likely. |
| **Attitudes towards teaching (ATTTEA)** | |
| ATTTEA01 | It would be good if the topic "the return of wolves to Germany" were taught in biology class. |
| ATTTEA02 | It would be good if the topic "the return of wolves to Germany" were promoted in school through education. |
| ATTTEA03 | It would be good if the topic "the return of wolves to Germany" were addressed in a project week. |
| **Enjoyment towards teaching (ENJ)** | |
| | *"If I imagine teaching the topic "the return of wolves" in biology class, I feel ..."* |
| ENJ01 | . . . happy. |
| ENJ02 | . . . joyful. |
| ENJ03 | . . . pleased. |
| **Perceived behavioral control towards teaching (PBC)** | |
| PBC01 | I could decide in my future biology class to teach about the return of the wolves to Germany. |
| PBC02 | I would have the possibility to teach about the return of wolves to Germany in my future school. |
| PBC03 | I would have the possibility to teach about the return of wolves to Germany during a project week. |

Note. * = Items were reversely coded for the analysis.

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
