# Peer review of "Do Pre-service Teachers Dance with Wolves? Subject-Specific Teacher Professional Development in A Recent Biodiversity Conservation Issue"

_sustainability, doi:10.3390/su11010047_

Reviewer 1 Report

This is a valuable and significant study which, while it is focused on wolf re-introduction education has broader implications as the authors have identified; including education of contentious issues, ESD and so on.  Whilst I am not an expert in this kind of quantitative statistical analysis I found the surveys to be well constructed and helpfully applied here (less detail on how they were changed would be fine). 

In terms of making improvements, my lack of expertise meant that I really struggled to follow the sections on pages 8 to 10 which described the data.  I wondered if that level of detail was necessary or if it could be included in an appendix for informing anyone who does this kind of research. However this is a decision for the authors because I suspect that anyone who does statistical work of this nature would probably find that section useful. 

If there is a major problem with this paper it lies in the way that the paper presents the purposes of education to be about saving wolves.  In fact the authors do refer to work (Vare and Scott) that points out that education is about learning not about inculcating sustainability but this happens very late in the text.  My recommendation would be that the paper foregrounds this work by introducing this dilemma right at the beginning and then keeping it as a critical thread throughout. 

In fact, the notion of changing values which the paper recommends could be criticised for being manipulative in its intent if the aim of changing values is to inculcate sustainable behaviour.  If the role of values education is to encourage teachers to take a critical stance to their own values and to enable children and adults to become aware of the role that values play in their decision making processes then this is an educational endeavor.  If the role of values education is to inculcate sustainable behaviour then this is not about learning but about behaviour change.  Whilst the outcome of behaviour change is in itself an important one if you work for an environmental charity it could be argued (and has been Scott and Gough, 2003) that it is not the role of schools to do this. 

This dilemma emerges in this paper and both sets of arguments are put forward, however, the paper is not clear about its purpose.  I think for this paper to make a strong case it needs to set out what that is at the start and then follow through, presenting both sides of the argument throughout but retaining a strong and clear positional statement on this dilemma.  Included in this change would be to reduce the way in which the authors make recommendations (page 16 paragraph 2 for example) which I think stretch a little further than the data support.

I would also prefer a clearer explanation of what closeness and psychological distance mean.

There are a few English language issues like the way in which the authors use the word: severe which need to be changed.

Change discussable on page 12 to debatable.

Author Response

Response to Reviewer 1 Comments

This is a valuable and significant study which, while it is focused on wolf re-introduction education has broader implications as the authors have identified; including education of contentious issues, ESD and so on.  Whilst I am not an expert in this kind of quantitative statistical analysis I found the surveys to be well constructed and helpfully applied here (less detail on how they were changed would be fine). 

Response 1: We thank the reviewer for the comment. We have some background in quantitative analysis and believe the details to be important for the replication of the results. If some of the content may still be too detailed, we are of course willing to transfer some content into the supplemental material on the demand of the editor.

In terms of making improvements, my lack of expertise meant that I really struggled to follow the sections on pages 8 to 10 which described the data.  I wondered if that level of detail was necessary or if it could be included in an appendix for informing anyone who does this kind of research. However this is a decision for the authors because I suspect that anyone who does statistical work of this nature would probably find that section useful. 

Response 2: Based on our quantitative background we included the information as needed for the replication of the analyses. We are of course also willing to transfer some content of this into a section in the supplemental material on the editor’s demand, but believe this would lead to a rather second paper focusing on methodological issues related to the main paper.

If there is a major problem with this paper it lies in the way that the paper presents the purposes of education to be about saving wolves.  In fact the authors do refer to work (Vare and Scott) that points out that education is about learning not about inculcating sustainability but this happens very late in the text.  My recommendation would be that the paper foregrounds this work by introducing this dilemma right at the beginning and then keeping it as a critical thread throughout. 

Response 3: This comment is very true and useful. We changed this and introduced the problem right in the beginning when discussing the issue of wolves and the problems of environmental issues (both page 2).

In fact, the notion of changing values which the paper recommends could be criticised for being manipulative in its intent if the aim of changing values is to inculcate sustainable behaviour.  If the role of values education is to encourage teachers to take a critical stance to their own values and to enable children and adults to become aware of the role that values play in their decision making processes then this is an educational endeavor.  If the role of values education is to inculcate sustainable behaviour then this is not about learning but about behaviour change.  Whilst the outcome of behaviour change is in itself an important one if you work for an environmental charity it could be argued (and has been Scott and Gough, 2003) that it is not the role of schools to do this. 

Response 4: We thank the reviewer for the clearer articulation of the two different approaches. Based on this we added further literature and clarified the problems of values from the beginning of the paper (especially page 2, line 84 and following). Overall, we did not intend value changes as an aim of education, as literature suggests artificial value changes may not be possible at all (for example the cited source of Manfredo et al., 2017). We nonetheless believe in the ability of education to be one contributor to positive attitudes and underlying value dimensions of students, as values lay the foundation of subsequent environmental behaviors. This is in accordance to prior studies in the literature, which explicitly include values as one part of ethical decisions and therefore also as an aim of ESD (for example the “Göttinger model”; Steffen & Hößle, 2014).

We further clarified this in the paper by adding literature, making clearer the differences between school students and preservice teachers, as well as explicitly stating teachers’ prohibition to indoctrinate students (“consensus of Beutelsbach”).

This dilemma emerges in this paper and both sets of arguments are put forward, however, the paper is not clear about its purpose.  I think for this paper to make a strong case it needs to set out what that is at the start and then follow through, presenting both sides of the argument throughout but retaining a strong and clear positional statement on this dilemma.  Included in this change would be to reduce the way in which the authors make recommendations (page 16 paragraph 2 for example) which I think stretch a little further than the data support.

Response 5: We believe the paper to have a clear purpose, based on the clear research question and specific hypotheses. But the mentioned recommendations indeed stretch the presented results, as we implied connections between variables we did not investigate in the paper (for example experiences as a contributor to affective learning outcomes).

We therefore thank the reviewer for this comment, changed this and clarified what the results support and what might be possible ways of addressing the affective dimensions based on other literature. We believe these recommendations to be important within the paper to articulate the consequences for practical considerations in teacher education.

I would also prefer a clearer explanation of what closeness and psychological distance mean.

Response 6: We initially used the word “closeness” as a more comprehensible word for the construct of psychological distance, as we evaluated this term to be rather technical and new to educational research. Furthermore, other studies did not apply the socio-psychological construct of psychological distance but the measured distance to the next wolf territory. While this may be a more objective measure, this underestimates the social dimension of the problem as described in the paper. We therefore selected “closeness” as a starting term, as this summarized both approaches of psychological distance as well as distance to the next wolf territory. 

We clarified this in the paper, but may also be convincible to use the term psychological distance from the beginning if the reviewer and editor recommends this as a way to prevent misunderstandings.

There are a few English language issues like the way in which the authors use the word: severe which need to be changed.

Change discussable on page 12 to debatable.

Response 7: Finally, we thank the reviewers for these suggestions and changed them in the manuscript.

Reviewer 2 Report

The paper is very well written, supplying a very good explanation of the work that the researchers carried out.

My only reservation is that it is a quantitative paper - based on quantifying attitudes. Personally as a researcher, I am under the impression that teachers often misreport their opinions on questionnaires. As the research did not include a control group, I recommend adding a table of quotations excerpted from interviews conducted with the teachers, to help convince the reader that these teachers' reported attitudes are genuine. If the research population is no longer available to the researchers for such interviews, this should not inhibit the paper's publication, though a disclaimer of such research limitations would be appropriate.

Author Response

Response to Reviewer 2 Comments

The paper is very well written, supplying a very good explanation of the work that the researchers carried out.

My only reservation is that it is a quantitative paper - based on quantifying attitudes. Personally as a researcher, I am under the impression that teachers often misreport their opinions on questionnaires.

Response 1: We thank the reviewer for this comment, which corresponds to our experience and is also documented in our paper for example with the increased means for some variables (e.g. attitudes towards the wolf).

While this points to a general problem of this kind of research and not only applies to teachers, the selected statistical methods partly account for this. First of all, correlations remain the same regardless if teachers report skewed positive attitudes, as the other variables are skewed in the same way. Furthermore, we selected robust statistical estimates to account for the non-normal distribution of the variables.

As the research did not include a control group, I recommend adding a table of quotations excerpted from interviews conducted with the teachers, to help convince the reader that these teachers' reported attitudes are genuine. If the research population is no longer available to the researchers for such interviews, this should not inhibit the paper's publication, though a disclaimer of such research limitations would be appropriate.

Response 2: While such interviews could be an interesting objective for a follow up study, similar to the questionnaires teachers will also be not really honest in interviews, as for example no biology teacher would fundamentally criticize sustainability at all based on his obligations to teach ESD. Therefore, the inclusion of interviews would only move, but not solve the problem. Furthermore, teachers may be even less honest in interviews compared to our questionnaires due to the guaranteed anonymity in the questionnaires and the direct social setting in interviews.

We nonetheless believe this as a significant problem and therefore added the recommended disclaimer concerning the interpretation of the results in the discussion (5.1.2.) and concluding section.

Reviewer 3 Report

The authors worked extensive on the literature in all sections, for representing the content and the problem, for establishing the methodology and also for interpreting their own results. The structure of methods section and results section follows a logical structure and all information needed is given, so that the reader is guided through the paper. I appreciate the possibility for getting insight into the original data, wich is included in the supplementary results.Overall, I really enjoyed reading!

In the following lines I list the minor issues. These should be corrected before publishing:

15 Canis lupus (capital C, written in italics for the scientific name)

17-20 This formulation is a bit misleading. "We found a higher motivation..." You do not give a hint about the comparison group. In the paper I learned, that you only made a comparative calculation from within the studied group. Your way of presenting the results in the abstract implies another outsider group.

57-59 Structure of sentence.

124 "... may play AN important ..."

348 "...recommended by Kline [105]..."

365 remove authors in [...], it should be [101,105]

388 Table 1: Maybe you could group the dependend variables in a graphical manner (like shading as you did in the Figure 1). This would help the reader to trace "personality traits 1-3", Closeness 4", and "motivation 5-7"

428 table 2 is diveded by page break

508 "implicate" - passive construction is not elegant to read at this position

577 same comment as 508, mybe "imply" is a better word for passive german "impliziert" - or "to implicate something" would be better

573 "relevanT processes"

582 "transferred -><- cautiously,"

584 "explained"

Author Response

Response to Reviewer 3 Comments

The authors worked extensive on the literature in all sections, for representing the content and the problem, for establishing the methodology and also for interpreting their own results. The structure of methods section and results section follows a logical structure and all information needed is given, so that the reader is guided through the paper. I appreciate the possibility for getting insight into the original data, wich is included in the supplementary results.Overall, I really enjoyed reading!

Response 1: Thank you very much for this comment. We are pleased that someone else enjoyed reading the paper as we were enjoying writing the paper.

In the following lines I list the minor issues. These should be corrected before publishing:

15 Canis lupus (capital C, written in italics for the scientific name)

17-20 This formulation is a bit misleading. "We found a higher motivation..." You do not give a hint about the comparison group. In the paper I learned, that you only made a comparative calculation from within the studied group. Your way of presenting the results in the abstract implies another outsider group.

57-59 Structure of sentence.

124 "... may play AN important ..."

348 "...recommended by Kline [105]..."

365 remove authors in [...], it should be [101,105]

388 Table 1: Maybe you could group the dependend variables in a graphical manner (like shading as you did in the Figure 1). This would help the reader to trace "personality traits 1-3", Closeness 4", and "motivation 5-7"

428 table 2 is diveded by page break

508 "implicate" - passive construction is not elegant to read at this position

577 same comment as 508, mybe "imply" is a better word for passive german "impliziert" - or "to implicate something" would be better

573 "relevanT processes"

582 "transferred -><- cautiously,"

584 "explained"

Response 2: We are especially thankful for these very detailed suggestions for minor language edits and included all of them in the revised manuscript.